# Interventions for Compassion Fatigue in Healthcare Providers—A Systematic Review of Randomised Controlled Trials

**DOI:** 10.3390/healthcare12020171

**Published:** 2024-01-11

**Authors:** Sanjay Patole, Dinesh Pawale, Chandra Rath

**Affiliations:** 1Neonatal Directorate, KEM Hospital for Women, Perth, WA 6008, Australia; dinesh.pawale@health.wa.gov.au (D.P.); chandra.rath@health.wa.gov.au (C.R.); 2School of Medicine, University of Western Australia, Perth, WA 6009, Australia

**Keywords:** compassion fatigue, burnout, secondary traumatic stress, professional quality of life

## Abstract

Background: Compassion fatigue is a significant issue considering its consequences including negative feelings, impaired cognition, and increased risk of long-term morbidities. We aimed to assess current evidence on the effects of interventions for compassion fatigue in healthcare providers (HCP). Methods: We used the Cochrane methodology for Systematic Reviews and Meta-Analyses (PRISMA) for conducting and reporting this review. Results: Fifteen RCTs (*n* = 1740) were included. The sample size of individual studies was small ranging from 23 to 605. There was significant heterogeneity in participant, intervention, control, and outcome characteristics. The tools for assessing intervention effects on compassion fatigue included ProQOL, compassion fatigue scale, and nurses compassion fatigue inventory. Thirteen out of the fifteen included RCTs had overall high risk of bias (ROB). Meta-analysis could not be performed given the significant heterogeneity. Conclusions: Current evidence on interventions for reducing compassion fatigue in HCPs is inadequate. Given the benefits reported in some of the included studies, well-designed and adequately powered RCTs are urgently needed.

## 1. Introduction

Empathy and compassion are at the core of the medical field, the business of providing care and comfort and, if possible, cure for patients. Empathy, the driver of prosocial behaviour, is the ability to understand and truly share the feelings of others [1,2,3]. Empathy is considered as the first step to more humane care [4] and critical for effective communication and healthcare [5,6,7,8,9,10].

Driven by empathy, compassion is the deep feeling that arises after becoming distressed by another person’s suffering, accompanied by a strong desire to help and improve his/her wellbeing [11,12,13,14]. Compassion leads to acts of kindness, taking empathy a step ahead [15,16,17,18].

Carrying the emotional burden of the suffering of other people over a long period can hurt the care provider due to the acute or cumulative effects of secondary stress [19,20]. Researchers hence consider empathy as a double-edged sword or the ‘cost of caring’ and emphasise the importance of awareness of this risk in healthcare providers [21,22,23,24,25,26]. The broad definition of compassion fatigue [27] is focused on secondary traumatic stress (STS), the emotional duress experienced by people in close contact with a trauma survivor [28,29]. Aspects of burnout were later added to this definition to include energy depletion as a well-known feature of compassion fatigue [29,30,31,32,33]. The secondary stress in compassion fatigue is similar to that in STS. However, compassion fatigue is characterised by reduced empathic capacity after repeated exposure to traumatic situations experienced by others [34,35]. Researchers suggest that STS occurs first and gives rise to compassion fatigue [36]. Compassion fatigue is a form of vicarious trauma with symptoms like that of post-traumatic stress disorder (PTSD) but without the inner change in worldview [37,38].

The significance of compassion fatigue cannot be overemphasised given its high prevalence [39,40,41,42]. Moreover, it has serious consequences including negative feelings, impaired cognition, and increased risk of long-term morbidities such as cardiovascular disease, and psychiatric illness [43,44,45].

Current evidence includes only a few systematic reviews in the field of compassion fatigue [46,47,48,49,50,51,52]. These reviews have significant heterogeneity in the participant (e.g., physicians, nurses, radiographers), faculty (e.g., oncology, emergency department), and location (e.g., rural, low-middle income nations) characteristics, and the evaluated aspects of compassion fatigue (e.g., prevalence, risk factors, interventions). Furthermore, some of these reviews focus on isolated components of compassion fatigue [48,52]. Interpreting and applying evidence from such heterogenous studies is hence difficult [53]. The recent systematic review by Chen et al. [54] has focused mainly on the effects of psychological interventions on empathy (not compassion) fatigue only in nurses. Similarly, the systematic review by Xie et al. [55] has assessed the effects of psychological interventions on compassion fatigue only in nurses.

Given the significance of underlining issues, we aimed to review the current evidence on the effect of various strategies for reducing compassion fatigue in healthcare providers. Our specific objective was to conduct a comprehensive and robust systematic review focused exclusively on randomised controlled trials (RCTs) of interventions for compassion fatigue in healthcare providers. Our findings will help in guiding research and clinical practice in the field.

## 2. Material and Methods

We followed the Cochrane methodology [56] and the Preferred Reporting Items for Systematic Reviews and Meta-Analyses (PRISMA) [57] statement for this systematic review. The protocol was approved by the university supervisor for this project [58].

### 2.1. Literature Search

PubMed, EMBASE, CINAHL, Emcare, and Cochrane library were searched from inception until December 2022. Grey literature was searched through Mednar (https://mednar.com (accessed on 10 January 2023)). Reviewers DP and SP independently searched the literature. The reference lists of the relevant studies were hand-searched to identify additional articles. No language restrictions were applied. The literature search focussed separately on systematic reviews and clinical/randomised trials assessing any intervention for compassion fatigue as follows:(1)The above-mentioned databases were searched for systematic reviews on interventions for compassion fatigue. PubMed were searched using the search terms (“compassion fatigue”[MeSH Terms] OR (“compassion”[All Fields] AND “fatigue”[All Fields]) OR “compassion fatigue”[All Fields]) AND (meta-analysis [Filter] OR systematic review [Filter]). Similar terms were used for other databases.(2)All databases were searched again for individual RCTs on interventions for compassion fatigue. PubMed was searched using the following broad key words: ((compassion fatigue) OR (compassion satisfaction)) OR (empathy fatigue) Filters: Clinical Trial, Randomized Controlled Trial. That automatic mapping system of PubMed expanded it to cover all the following terms: (“compassion fatigue”[MeSH Terms] OR (“compassion”[All Fields] AND “fatigue”[All Fields]) OR “compassion fatigue”[All Fields] OR ((“empathy”[MeSH Terms] OR “empathy”[All Fields] OR “compassion”[All Fields]) AND (“personal satisfaction”[MeSH Terms] OR (“personal”[All Fields] AND “satisfaction”[All Fields]) OR “personal satisfaction”[All Fields] OR “satisfaction”[All Fields] OR “satisfactions”[All Fields] OR “satisfaction s”[All Fields])) OR ((“empathy”[MeSH Terms] OR “empathy”[All Fields]) AND (“fatiguability”[All Fields] OR “fatiguable”[All Fields] OR “fatigue”[MeSH Terms] OR “fatigue”[All Fields] OR “fatigued”[All Fields] OR “fatigues”[All Fields] OR “fatiguing”[All Fields] OR “fatigueability”[All Fields]))) AND (clinicaltrial[Filter] OR randomizedcontrolledtrial[Filter]). Similar terms were used for other databases.

### 2.2. Inclusion Criteria

RCTs and quasi-RCTs assessing interventions for compassion fatigue were eligible for inclusion. Studies not assessing interventions for compassion fatigue or dealing with STS, burnout, and vicarious trauma were excluded. We excluded reviews, editorials, case reports, letters and commentaries but read them to identify potential studies.

The primary outcome was the effect of the intervention on compassion fatigue (Mean pre-post difference in scores) measured by a validated tool such as the Professional Quality of Life (ProQOL) scale [59,60,61,62]. Secondary outcomes included effects of the intervention on individual components of compassion fatigue, including burnout and STS, and other outcomes reported in the individual studies.

### 2.3. Assessing the Risk of Bias (ROB)

The Cochrane ROB tool (RoB 2.0) was used to assess the ROB based on the following criteria [61]: 1. Was the allocation sequence adequately generated? 2. Was allocation adequately concealed? 3. Was blinding of participants, personnel, and outcome assessors adequate? 4. Were incomplete outcome data adequately addressed? 5. Are reports of the study free of selective outcome reporting? 6. Was the study apparently free of other problems? Overall, this tool assesses the randomisation process, deviations from intended interventions, missing outcomes, the method for measuring the outcomes and selective reporting [63].

### 2.4. Data Extraction

Two reviewers (SP and DP) independently extracted the data using a prespecified data collection form. All reviewers verified the information about study design and outcomes independently. Disagreements were resolved through joint discussions if required. If required, we planned to contact the authors of included studies for incomplete data or clarifications. Studies were excluded after two failed attempts to contact the authors.

### 2.5. Data Synthesis

We planned to conduct the meta-analysis using the Review manager V.5.4 (Cochrane collaboration, Nordic Cochrane Centre, Copenhagen, Denmark). We selected the random-effects model for meta-analysis given the significant heterogeneity [64]. We chose relative risk (RR) size and the mean difference (MD) to express the effects for dichotomous and continuous outcomes, respectively.

### 2.6. Assessment of Heterogeneity, Publication Bias and Certainty of the Evidence (CoE)

We planned to assess statistical heterogeneity with the χ^2^ test, I^2^ statistic and by visual inspection of the forest plot (overlap of CIs) [65]. As recommended, an I^2^ value of >50% indicates substantial heterogeneity [65]. When pooling was not possible, we planned to present the non-pooled data in tabular form. We planned to check publication bias by a funnel plot if more than 10 RCTs were available for meta-analysis. We also planned to assess publication bias objectively by Egger’s test [66,67]. We planned to report the certainty of evidence using the GRADE framework [68].

### 2.7. Exploring Heterogeneity

Heterogeneity was explored by considering the variations in participant, interventions, comparator, and outcomes in included studies. We planned to explore differences in the design (e.g., randomised vs. quasi-randomised) and settings (ICU vs. non-ICU, urban vs. rural-remote) of the included studies.

## 3. Results

The literature search retrieved 865 potentially relevant citations. After carefully reviewing the abstracts, 260 duplicate studies were excluded. A total of 570 studies were excluded after reading titles and abstracts. Twenty articles were excluded after reading 35 full texts. Finally, 15 RCTs were included in the review [69,70,71,72,73,74,75,76,77,78,79,80,81,82,83]. The flow diagram of the study selection process is shown in Figure 1.

The characteristics of the 15 included studies are described in Table 1. There was significant heterogeneity in the participants, interventions, control, and outcome characteristics, and the settings, and timing of assessing the effect of the intervention in these studies. The sample size of individual studies was small, ranging from 23 to 605 participants [69,70,71,72,73,74,75,76,77,78,79,80,81,82,83]. The total sample size was 1740 (mean: 116). Nine out of fifteen included studies showed reduction in either compassion fatigue or its components after intervention [70,71,73,74,75,77,79,81,83]. Six studies did not show any effect of intervention on compassion fatigue or its components [69,72,76,78,80,82]. The design of a quasi-experimental study with ‘nurses from a similar unit’ as control group was unclear [82].

Majority of the included studies (11/15) recruited nurses working in different settings ranging from intensive care unit, and emergency department to elderly care centres, and trauma centres. Two studies recruited mental health workers, behavioural health clinicians, and another recruited a proportion (39.5% of 253) of participants as psychologists, psychotherapists, coaches, and counsellors. The studies were conducted in various nations including Iran (*n* = 3), Turkey (*n* = 3), Slovakia, Israel, Spain, Korea (*n* = 1 each), and USA (*n* = 5).

The interventions ranged from motivational messages, acupressure, and guided imagery to inhalation of patchouli oil, chromotherapy, and Emotion-Focused Training for Health Professionals (EFT-HP). The various tools for assessing intervention effects included the ProQOL, compassion fatigue scale, nurses compassion fatigue inventory, Disaster-Helper Self-Efficacy Scale (DHSES), Rosenberg self-esteem scale (RSE), Self-rated Hope Scale, Mastery scale, and the Pittsburgh Sleep Quality Index (PSQI).

Four studies had ‘wait list’ controls [71,72,79,81]. Other studies did not offer the intervention to participants in the control group.

Trial registration was reported in 9/15 included RCTs [71,73,74,75,76,80,81,82,83]. An approach to sample size and power estimation was reported in 6/15 included studies [69,70,72,77,78,80]. Some of these trials did not report the rationale for sample size estimation adequately [72,78]. The distinction between primary and secondary outcomes was unclear in majority of the included RCTs. Five studies were reported as a pilot [73,74,80,81,82]. The tools used in the included studies for assessing the effect of interventions specifically on compassion fatigue included ProQOL [84,85,86,87,88,89], compassion fatigue scale [70], and nurses compassion fatigue inventory [69,90].

Our literature search revealed two recently published systematic reviews [54,55]. Chen et al. included 7 RCTs (Total 513 nurses) and assessed the effects of psychological intervention on empathy fatigue in nurses. Four of the seven RCTs included in Chen review were identified in our literature search and remaining three were judged ineligible being non-RCTs [91,92,93]. Xie Wanging et al. that assessed the effects of psychological interventions on compassion fatigue among nursing staff was excluded as the full paper was not available despite contacting the authors [55]. The translated abstract in English was unreliable (e.g., “Thirteen RCTs involving 13 nursing staff were included”).

### 3.1. Approach to Synthesis

Given the significant heterogeneity in participants, interventions (type and duration), controls, and outcomes as well as the settings, tools for assessments, and timing of assessments post-intervention in the included RCTs pooling of data was considered inappropriate. Hence, a narrative approach to synthesis was undertaken. A funnel plot to assess the probability of publication bias was not considered.

### 3.2. Risk of Bias Assessment (Figure 2)

Thirteen out of the fifteen (87%) included studies had overall high ROB. The two domains with high ROB involved the randomization process (12/15 (80%) included studies) and in the measurement of outcomes (13/15 (87%) of the 15 included studies). The high ROB in the measurement of outcomes domain is because of the nature of intervention and reporting of outcomes by the participants themselves. Figure 2 provides the results of the ROB assessment.

**Figure 2 healthcare-12-00171-f002:**
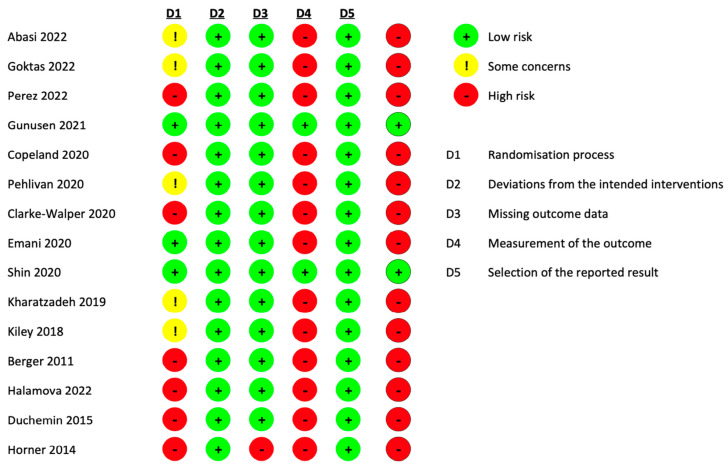
Risk of bias assessment of the included studies [69,70,71,72,73,74,75,76,77,78,79,80,81,82,83].

## 4. Discussion

Our systematic review revealed 15 RCTs (*n* = 1740) assessing various interventions for reducing compassion fatigue in healthcare providers from different sectors. Most studies used the ProQOL scale for the purpose. Significant heterogeneity in characteristics of the settings, participants, interventions, comparator groups, outcomes, and timing (and in few studies, the tool) for assessing the effects of the intervention meant conducting meta-analysis was inappropriate. All included studies had small sample size and high risk of bias. Deriving a robust conclusion from such inadequate and low-quality data was hence difficult. Our findings indicate the urgent need for well-designed and adequately powered definitive RCTs of interventions to reduce the risk of compassion fatigue in healthcare providers.

The significance of the methodology and results of two systematic reviews [46,54] need to be discussed in the context of our systematic review. Cocker et al. [46] reported a systematic review of studies assessing psychological interventions for reducing compassion fatigue in healthcare providers in various settings. The studies used validated measures such as Pro-QoL, Maslach Burnout Scale [94], or the Resilience Scale [95]. Of the total 229 studies identified, only 13 studies were included in the review (*n* = 671, range: 7 to 153). There was significant heterogeneity in participant, setting, and intervention characteristics, the period of follow up, and the study design including the type of control group. Participants were not truly representative of the target population as their selection was based on convenience rather than random sampling. Given the small total sample size, and the fact that 11 of the 13 included studies were non-RCTs it is difficult to derive clear and valid conclusions from this systematic review. The results of the included two small RCTs are also not helpful in this context [46]. The ideal intervention for compassion fatigue would be one that reduces burnout and STS (the negative aspect) and increases compassion satisfaction, the positive aspect of professional quality of life. One of the thirteen included studies that showed such results involved a two-step intervention amongst emergency nurses [96].

Chen et al. reported a systematic review assessing the effects of psychological interventions on empathy (not compassion) fatigue in nurses [54]. Meta-analysis of data from 7 included RCTs (*n* = 513 nurses) showed that empathy fatigue (SMD = −0.22, 95% CI: −0.42~−0.02, *p* = 0.03) and burnout (SMD = −0.37, 95% CI: −0.56~−0.19, *p* < 0.001) scores were significantly lower in the intervention vs. control group. The empathy satisfaction score was significantly higher in intervention vs. control group (SMD = 0.45, 95% CI: 0.27–0.63, *p* < 0.001). The effects of the intervention were significantly different in different settings including ICU, paediatric, and other departments (*p* = 0.0007). Both mindfulness (SMD = 0.50, 95% CI: 0.24–0.77, *p* = 0.0002) and other interventions (SMD = 0.41, 95% CI: 0.16–0.65, *p* = 0.001) had a significant effect on empathy satisfaction between the intervention vs. control group [54]. We believe that pooling data from heterogeneous interventions under a common label as ‘psychological’ interventions is debatable.

Overall, the data on the range of interventions from studies included in our systematic review and those by Cocker et al. [46], Chen et al. [54], and the findings of some of the studies discussed here [97,98,99] should be useful in selecting intervention for assessing in future studies.

Selecting the design of future studies in this field is a critical issue. RCTs are considered as the gold standard for clinical research to assess the effects (benefits and risks) of an intervention [100]. The strength of the RCT design lies in its ability to control for not only the known, but also the unknown confounders. No other study design can achieve this goal [101,102]. However, well-designed large adequately powered RCTs are not easy to conduct considering the difficulties related to their feasibility, acceptability, and need for resources, and expertise. A systematic review and meta-analysis of RCTs with comparable characteristics of participants, interventions, comparator, and outcomes provides a robust way to synthesise the evidence with more power and precision when the available RCTs are small and have conflicting results [103,104].

Randomisation at individual participant level will be difficult in RCTs of interventions to reduce compassion fatigue and/or improve compassion satisfaction, especially in small organisations. Contamination of the control group participants is inevitable if they work in the same section of an organisation as those assigned to the intervention group. This is particularly a possibility given the nature of psychological interventions and the fact that humans influence others consciously or subconsciously. Cluster RCT design is hence more appropriate for such studies [105,106,107]. The limitations of cluster RCTs include the need for biostatisticians with expertise (e.g., power, precision, number and size of the clusters, imbalance between clusters, intra-cluster correlation coefficient) [108,109,110,111,112,113,114,115]. Furthermore, the power and precision of results generated by a cluster-RCT are lower compared to those from a conventional RCT. The stepped wedge design is an alternative for rigorous evaluation of organisational interventions such as those for managing compassion fatigue [116,117]. This unique design provides an option where all trial participants receive the intervention but the order in which the intervention is received is randomised.

As for interventions, considering their simplicity, acceptability, ease of application, and logistics of implementation, as well as resources and socioeconomic and cultural differences is important in designing trials in this field. The ProQOL scale is perhaps the most appropriate tool for assessing effects of interventions for compassion fatigue considering it is culturally adapted and validated in several countries and has high reliability and validity [118,119]. Biological markers such as salivary amylase will optimise the validity of the results RCTs [81]. Salivary amylase is a marker of stress-induced activation of the autonomic nervous system. Its secretion is controlled by sympathetic activation through β-adrenergic receptors. It has been used as a marker to measure effects of stress reduction interventions [81]. The first responders and those in an intensive care set-up could be the priority population for future trials given they are at high risk of compassion fatigue. Equally important will be the considerations such as the settings (high- vs. low-income counties), language, and culture. Considering the ethical implications for any intervention to be assessed is an important issue in the design of future RCTs.

## 5. Conclusions

Our comprehensive systematic review with robust methodology including assessment of the risk of bias by a validated toll (ROB-2) indicates that current evidence on interventions for reducing compassion fatigue in healthcare providers is inadequate. Given the benefits reported in some of the included studies, there is an urgent need for well-designed and adequately powered definitive RCTs in this field. Our systematic review is helpful in the designing such trials.

## Figures and Tables

**Figure 1 healthcare-12-00171-f001:**
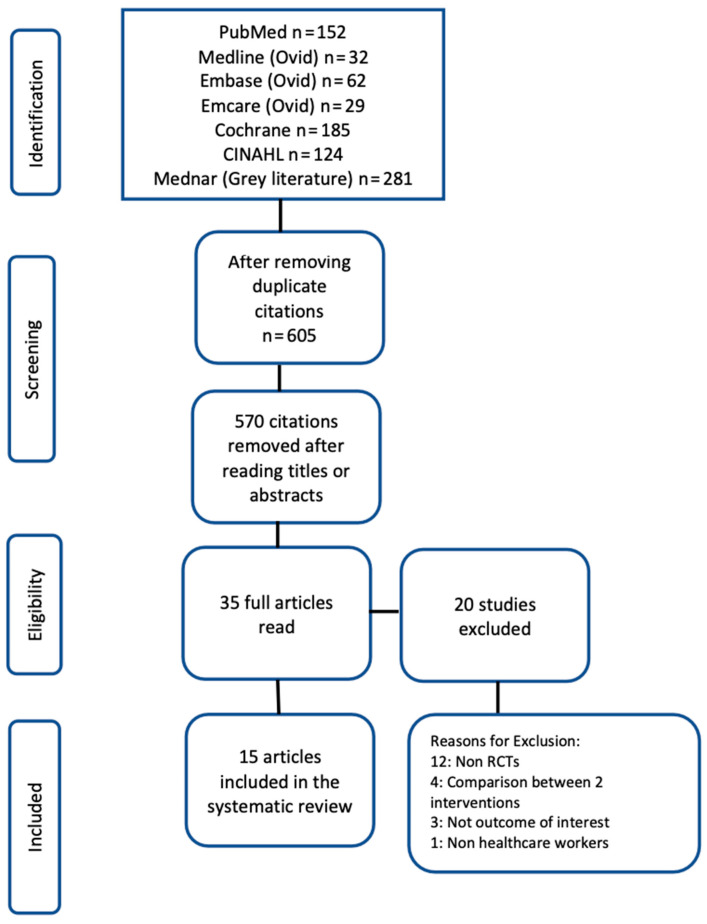
PRISMA flow chart for study selection.

**Table 1 healthcare-12-00171-t001:** Characteristics of included studies.

No	Study Details (Ref)	Participants (*n*)	Intervention	Control	Outcomes	Tool of Assessment
1	Abasi 2022, Iran [69]	Nurses in ED and ICU (80)	Acupressure on the Shenmen point (1 month)	No acupressure	CF (⬄) STS Sleep quality	NCFI, PSQI
2	Goktas 2022, Turkey [70]	Nurses working in ED (65)	Motivational messages (21 days)	No motivational messages	CF (⇩) Communication skills Job Satisfaction	CFS, Job satisfaction scale, communication skills scale.
3	Perez 2022, Spain [71]	Nurses working in elderly care centre (74)	MBCT program (6 weeks)	No MBCT for 3-month study period (Wait list)	CS (⬄) CF (⇩) BO (⇩)	ProQoL (R-IV)
4	Gunusen 2022, Turkey [72]	Nurses in general inpatient units (48)	Nurse-led program based on CBT (4 weeks)	No CBT for 6-month study period (Wait list)	CS (⬄) CF (⬄) BO (⬄) Psychological distress	ProQoL (R-IV)
5	Halamova 2022, Slovakia [73]	Healthcare and allied professionals (253)	EFT-HP (2 weeks)	No EFT-HP	BO (⇩) STS (⇩) self-criticism (⇩) self-compassion (⇧) compassion for others (⇧)	STS scale, OLBI, FSCRS, SOCS-S, SOCS-O
6	Copeland 2020, USA [74]	Nurses in level 1 trauma centre (23)	Meditation, Outing, Gratitude and Journaling (6 weeks)	No change in work practice for 6-week study period	CS (⇧) STS (⇩) BO (⇩)	ProQoL
7	Pehlivan 2020, Turkey [75]	Nurses in oncology haematology inpatient services, outpatient chemotherapy units, and bone marrow transplant units (125)	CFRP (5 weeks)	No CFRP	CS (⇧) CF (⇩) BO (⇩) Perceived stress Resilience	ProQoL (R-IV), Perceived stress scale, Resilience scale
8	Clarke-Walper 2020, USA [76]	Behavioural health clinicians in the U.S. Department of Veterans Affairs, Department of Défense, and general community (605)	PTSD Clinicians Exchange (12 months)	Newsletter only	CS (⬄) STS(⬄) BO (⬄) Practice patterns EBPAS Organisational support (7-point scale)	PTSD Clinicians Exchange that included ProQoL (R-V), Questions about clinical practice, EBPAS, new 7-point scale developed for assessing organisational support.
9	Emani 2020, Iran [77]	Nurses working in ICU (80)	Chromotherapy with educational sessions and individualised consulting sessions (3 months)	No chromotherapy, education or consulting	CS (⇧) STS (⇩) BO (⇩)	ProQoL
10	Shin 2020, Korea [78]	Nurses working in emergency department (60)	Short-term inhalation of Patchouli oil (24 h)	Pure sweet almond oil inhalation	CS (⇧) CF (⬄) BO (⬄) Stress levels Blood pressure	ProQoL (R-V), visual analogues scale for stress, an electronic sphygmomanometer
11	Kharatzadeh 2019, Iran [79]	Nurses in the intensive and critical care units (60)	ERT sessions (six sessions × 2 h each)	No ERT sessions for the study period (Wait list)	CS (⇧) CF (⬄) BO (⇩) Cognitive Emotion Regulation Depression, Anxiety and Stress	ProQoL (R-V), Cognitive Emotion Regulation Questionnaire, Depression, Anxiety and Stress Scale
12	Kiley 2018, USA [80]	Mental health workers (69)	Guided imagery as a relaxation technique (4 weeks)	No guided imagery	CS (⬄) STS (⬄) BO (⬄) Perceived Stress Sleep Quality Anxiety	ProQoL (R-V), Perceived Stress Scale, PSQI, State Trait Anxiety Inventory
13	Duchemin 2015, USA [81]	Personnel working in the SICU (32)	Group MBI (8 weeks)	No group mindfulness-based intervention (MBI) (wait list)	BO (⇩) Salivary α-amylase (⇩) Psychological stress (⇩)	ProQOF, Maslach Burnout Inventory, PSS, DASS-21
14	Horner 2014, USA [82]	Personnel working Medical–Surgical units providing intermediate intensity of care (86)	Mindfulness training program (10 weeks)	No mindfulness training program	CS (⬄) BO (⬄) Individual stress (⬄) Unit stress (⬄) Mindful Attention Awareness (⬄) Patient satisfaction (⬄)	ProQoL (R-V), MAAS, HCAHPS survey
15	Berger 2011, Israel [83]	Nurses in Well baby clinic (80)	Sessions on psycho-educational knowledge and stress management techniques (12 weeks)	No sessions on psycho-educational knowledge and stress management techniques till study end (Wait list)	5. Self-efficacy 6. Secondary traumatization: CS (⇧), CF (⇩), and BO (⇩) 7. Self-esteem 8. Hope 9. Sense of mastery	DHSES, ProQoL, RSE, Self-rated Hope Scale, Mastery scale.

ED: Emergency department; ICU: Intensive care unit; CF: Compassion fatigue; CS: Compassion satisfaction; BO: Burnout; STS: secondary traumatic stress; NCFI: Nurses’ Compassion Fatigue Inventory; PSQI: Pittsburgh Sleep Quality Index; MBCT: Mindfulness-Based Cognitive Therapy; CBT: Cognitive behaviour therapy; CFS: Compassion Fatigue Scale; ProQoL: Professional Quality of Life Scale; OLBI: Oldenburg burnout inventory; EFT-HP: Emotion-focused training for helping professionals; FSCRS: Forms of self-criticizing/attacking and self-reassuring scale; SOCS-S: Sussex-Oxford compassion for the self-scale; SOCS-O: Sussex-Oxford compassion for others scale; CFRP: Compassion fatigue resiliency program; PTSD: Post traumatic stress disorder; EBPAS: Evidence-Based Practice Attitudes Scale; ERT: Emotion regulation training; ProQOF: Professional Quality of Life; MBT: Mindfulness-based intervention; PSS: Perceived Stress Scale; DASS-21: Depression Anxiety Stress Scale; SICU: surgical intensive care unit; MASS: Mindful Attention Awareness Scale; HCAHPS: Hospital Consumer Assessment of Healthcare Providers and Systems; DHSES: Disaster-Helper Self-Efficacy Scale; RSE: Rosenberg self-esteem scale. ⇧—Increase; ⇩—decrease; ⬄—No change.

## Data Availability

Not applicable.

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
