# Peer review of "Interventions for Compassion Fatigue in Healthcare Providers—A Systematic Review of Randomised Controlled Trials"

_healthcare, 2024, doi:10.3390/healthcare12020171_

Round 1
Reviewer 1 Report
Comments and Suggestions for Authors
The main problem identified in this review is the lack of clarity in knowledge advancement when comparing results with the other two reviews (Chen et al., 2022; Xie et al., 2022). I think these reviews could have supported the presentation of the research problem and its justification, but not polarized the discussion. Therefore, it is unclear to what extent knowledge has advanced, especially to support practice and decision-making. I think the review mentioned above by Chen et al., 2022 would already be useful to identify the gap in this area and serve as a basis for formulating new research questions.
The authors do not provide information about the registration of the protocol, only that it was approved by the supervisor.
Other questions: adjust the tense on page 2, line 73. Were any studies excluded after unsuccessful attempts to contact authors? What was the time frame of the search? Why were other databases such as Web of Science and Scopus not investigated? Which reference manager was used? How did the removal of duplicates happen? Also manually?
In the results, the authors mentioned having included 4 RCTs from the review by Chen et al., but later in the discussion, they changed it to 5. The authors could have further explored the reason for the high risk of bias assessments in the ROB-2 of each study. Specifically, as per the comment above, on page 9, line 251, it is unclear what the review adds to the two existing ones regarding the selection of interventions for evaluation in future studies. Furthermore, these interventions were not properly discussed.
Author Response
1) Comment: The main problem identified in this review is the lack of clarity in knowledge advancement when comparing results with the other two reviews (Chen et al., 2022; Xie et al., 2022). I think these reviews could have supported the presentation of the research problem and its justification, but not polarized the discussion. Therefore, it is unclear to what extent knowledge has advanced, especially to support practice and decision-making. I think the review mentioned above by Chen et al., 2022 would already be useful to identify the gap in this area and serve as a basis for formulating new research questions.
Response: Thanks for the comment. Chen et al. and Xie et al. have now been mentioned in the “Introduction” section, and their limitations are covered briefly. This justifies the need for our systematic review. We have added the following text in the introduction page 2 lines 64-70:
“The recent systematic review by Chen et al 2022(54) has focused mainly on the effects of psychological interventions on empathy (not compassion) fatigue only in nurses. Similarly, the systematic review by Xie et al, 2022 (55) has assessed the effects of psychological interventions on compassion fatigue only in nurses.”
2) Comment: The authors do not provide information about the registration of the protocol, only that it was approved by the supervisor.
Response: The protocol for this leadership research project was registered with Deakin University, Melbourne, Australia, as a part of master’s degree in leadership. The link to the Deakin University Supervisor is provided in the manuscript under 'Registration’. The review was conducted only after approval of the protocol by the supervisor to assure there was no violation of the protocol.
3) Comment: Other questions: adjust the tense on page 2, line 73. Were any studies excluded after unsuccessful attempts to contact authors? What was the time frame of the search? Why were other databases such as Web of Science and Scopus not investigated? Which reference manager was used? How did the removal of duplicates happen? Also manually?
Response: Thanks for the queries. The tense has now been changed as suggested. We used EndNote X9 reference manager for managing citations. Duplicates were removed using EndNote citation manager and details of which are mentioned in Figure 1. None of the studies were excluded after unsuccessful attempts to contact authors. We contacted Xie et al for the full manuscript of their systematic review, but the message failed to deliver. Databases were searched from inception until December 2022, as mentioned on page 2, line 74. The Cochrane collaboration recommends searching at least 3 databases while conducting a systematic review of interventions (1). In addition to the 3 recommended databases, we have searched many other databases including Grey literature to make the search methodology robust. We also hand searched the relevant articles for additional studies. Furthermore, search of the database Psych info revealed 8 studies which were already identified in other databases. In this context we believe searching additional databases was not necessary.
- Higgins JPT, Thomas J, Chandler J, Cumpston M, Li T, Page MJ, Welch VA (editors).Cochrane Handbook for Systematic Reviews of Interventionsversion 6.4 (updated August 2023). Cochrane, 2023. Available from www.training.cochrane.org/handbook.
4) Comment: In the results, the authors mentioned having included 4 RCTs from the review by Chen et al., but later in the discussion, they changed it to 5. The authors could have further explored the reason for the high risk of bias assessments in the ROB-2 of each study. Specifically, as per the comment above, on page 9, line 251, it is unclear what the review adds to the two existing ones regarding the selection of interventions for evaluation in future studies. Furthermore, these interventions were not properly discussed
Response: We included 4 studies from Chen et al (The other 3 were not RCTs). We have addressed this issue in the revised manuscript. Page 8 lines 203 and 204:
“Our systematic review includes only 4 of the 7 studies in Chen et al as on close scrutiny, the other three (93-95) were not found to be RCTs”
The risk of bias assessment for the studies included from Chen et al is already provided in Figure 2 and the reason for the heterogeneity is shown in Table 1. As mentioned earlier, the full text of the systematic review is not available even after contacting the authors.
The systematic review by Xie et al assessed the prevalence of compassion fatigue and compassion satisfaction in nurses. They included only the studies which used the ProQOL scale for measurement of the outcomes of interest. In contrast our systematic review was focussed on effects of any interventions to reduce compassion fatigue in healthcare professionals.
Chen et. al. 2022 included 7 studies which used ProQOL scale as a measurement tool. However, their focus was on empathy fatigue in nurses. In contrast our search methodology was robust and identified 6 additional studies which reported compassion fatigue score using the ProQOL tool. As mentioned earlier, our SR is comprehensive in assessing the effects of any intervention to reduce compassion fatigue in healthcare providers in general.
As for the ROB in the studies included in our systematic review, we have covered variations in the participant, intervention (e.g., type, intensity, duration), comparator and outcome characteristics as well as the settings, and time period (Table 1, Figure 2). Covering these details in text is difficult given the limitation on word limit and the risk of duplication.
Given the inadequate evidence and significant heterogeneity it is difficult to suggest options for interventions suitable for testing large RCTs.
Reviewer 2 Report
Comments and Suggestions for Authors
This is a very interesting and valuable review. It is heavily focussed on RECTs and it is unclear why more qualitative elements were discounted (ie in the introduction the emotional aspects are alluded to but the interventions do not address the emotional impact). That said the paper is very focussed and this does not detract from its importance. It would be useful to know the range of healthcare workers this applies to as the role and responsibility may be very different (ie nurses or location ie trauma) and issues such as locus of control etc. The data is well presented but the analysis could have a little more depth ie the range of countries form the literature is interesting is this a consideration ie culture?. Overall very interesting.
Author Response
Comment: This is a very interesting and valuable review. It is heavily focussed on RCTs, and it is unclear why more qualitative elements were discounted (ie in the introduction the emotional aspects are alluded to but the interventions do not address the emotional impact). That said the paper is very focussed and this does not detract from its importance. It would be useful to know the range of healthcare workers this applies to as the role and responsibility may be very different (ie nurses or location ie trauma) and issues such as locus of control etc. The data is well presented but the analysis could have a little more depth ie the range of countries form the literature is interesting is this a consideration ie culture? Overall, very interesting.
Response: Thanks for the comments. We agree that the evaluation of the qualitative aspect is an important issue. However, covering this aspect was beyond the scope of our systematic review. Moreover, in RCTs, the focus is on the primary outcome of the included studies. Table 1 does cover the details on the setting, country, and participants. The significance of variation in participant characteristics, country (differences in language and culture), study setting (e.g., ward, ICU), and time period have been covered in the “Results” (Page 8 Line 178-190), and the “Discussion” (page 10, line 235-243) section of the manuscript.
Reviewer 3 Report
Comments and Suggestions for Authors
Dear Authors, please find my comments below
Abstract:
- Could you briefly highlight specific findings or trends observed with assessment tools used for compassion fatigue?
- Add a notable finding or trend from the RCTs to increase reader engagement.
Introduction:
- Could you streamline the introduction slightly for brevity?
- Could you provide a more precise transition to the specific focus of the systematic review?
- Could you include a brief preview or outline of the main points?
Methods:
- Could you explain why you chose specific databases or sources for the literature search?
- Could you provide brief details on how heterogeneity will be assessed and addressed in the analysis?
- Could you explain why publication bias is a concern and its potential implications?
Results:
- Could you explain why you chose a narrative synthesis approach?
- Provide a summary or discussion of key findings from Table 1.
- Discuss the implications of the high risk of bias observed in most studies.
Discussion:
- Touch upon potential solutions or strategies to overcome challenges in conducting RCTs.
- Could you include practical recommendations or considerations for researchers planning future RCTs in this area?
Conclusion:
- Add a separate conclusion summarizing essential findings and providing a clear overall research assessment.
Comments on the Quality of English Language
There are issues with grammar and language that need to be addressed. You should thoroughly proofread and edit the paper to improve the overall quality of writing.
Author Response
Abstract:
1) Comment: Could you briefly highlight specific findings or trends observed with assessment tools used for compassion fatigue?
Response: Thanks for your comments. Our systematic review focussed only on RCTs of any intervention on compassion fatigue in healthcare providers. Meta-analysis was not possible given the significant heterogeneity in almost all key aspects of included RCTs, hence providing a trend is difficult. However, we have modified the conclusion of the abstract to include a trend observed in the findings as follows:
“Given the benefits reported in some of the included studies well-designed adequately powered RCTs are urgently needed”
2) Comment: Add a notable finding or trend from the RCTs to increase reader engagement.
Response: Please see the response above.
Introduction:
3) Comment: Could you streamline the introduction slightly for brevity?
Response: The ’Introduction’ section has been edited to reduce its word count from 584 to 481.
4) Comment: Could you provide a more precise transition to the specific focus of the systematic review?
Response: We have addressed this issue by streamlining the ‘Introduction’ as suggested.
5) Comment: Could you include a brief preview or outline of the main points?
Response: We have justified the main aim of the systematic review by providing information on other recently published systematic reviews.
Methods:
6) Comment: Could you explain why you chose specific databases or sources for the literature search?
Response: The Cochrane collaboration recommends searching at least 3 databases (CENTRAL, Medline, Embase) while conducting a systematic review of interventions (1). In addition to the 3 recommended databases, we have searched many other databases including Grey literature to make the search methodology robust. We also hand searched the relevant articles for additional studies. Furthermore, search of the database Psych info revealed 8 studies which were already identified in other databases. In this context we believe searching additional databases was not necessary.
- Higgins JPT, Thomas J, Chandler J, Cumpston M, Li T, Page MJ, Welch VA (editors).Cochrane Handbook for Systematic Reviews of Interventionsversion 6.4 (updated August 2023). Cochrane, 2023. Available from www.training.cochrane.org/handbook.
7) Comment: Could you provide brief details on how heterogeneity will be assessed and addressed in the analysis?
Response: Exploration of heterogeneity involved careful consideration of the differences in type of participants, interventions (type and duration), controls, and outcomes as well as the settings, tools for assessments, and timing of the post-intervention assessments in the included RCTs as shown in table 1. Please refer to point 2.7 (pages 3 and 4)
8) Comment: Could you explain why publication bias is a concern and its potential implications?
Response: Though planned, we could not assess publication bias as meta-analysis could not be conducted given the significant heterogeneity in almost all key characteristics of the included RCTs.
The validity of systematic reviews can be compromised by publication bias, which occurs when the publication or non-publication of research findings is determined by the direction or strength of the evidence, and by outcome reporting bias whereby only a subset of outcomes, typically those most favourable, are reported (1,2).
1.Page MJ, Higgins JPT, Sterne JAC. Chapter 13: Assessing risk of bias due to missing results in a synthesis. In: Higgins JPT, Thomas J, Chandler J, Cumpston M, Li T, Page MJ, Welch VA (editors). Cochrane Handbook for Systematic Reviews of Interventions version 6.4 (updated August 2023). Cochrane, 2023. Available from www.training.cochrane.org/handbook.
- Nair AS. Publication bias-Importance of studies with negative results!. Indian Journal of Anaesthesia. 2019 Jun;63(6):505.
Results:
9) Comment: Could you explain why you chose a narrative synthesis approach?
Response: Pooling of data was considered in-appropriate given the significant heterogeneity in participants, interventions (type and duration), controls, and outcomes as well as the settings, tools for assessments, and timing of post-intervention assessments in the included RCTs. Hence, a narrative approach to synthesis was undertaken (Page 9, lines 209-214).
10) Comment: Provide a summary or discussion of key findings from Table 1.
Response: We have covered this information adequately in the three paragraphs on page 8 and 9 in the “Results” section.
11) Comment: Discuss the implications of the high risk of bias observed in most studies.
Response: The presence of high risk of bias in key domains of the included RCTs questions the validity of their results, making the evidence unreliable (1). The variation in PICO, design and setting are covered well in table 1, figure 2. Hence discussion of those differences in the result section in text would be duplication.
- Higgins JPT, Savović J, Page MJ, Elbers RG, Sterne JAC. Chapter 8: Assessing risk of bias in a randomized trial. In: Higgins JPT, Thomas J, Chandler J, Cumpston M, Li T, Page MJ, Welch VA (editors). Cochrane Handbook for Systematic Reviews of Interventions version 6.4 (updated August 2023). Cochrane, 2023. Available from training.cochrane.org/handbook.
Discussion:
12) Comment: Touch upon potential solutions or strategies to overcome challenges in conducting RCTs.
Response: This issue has been covered adequately in the three paragraphs in discussion (Pages 10, 11).
13) Comment: Could you include practical recommendations or considerations for researchers planning future RCTs in this area?
Response: This issue has been covered adequately in the three paragraphs in discussion (Pages 10, 11).
Conclusion:
14) Comment: Add a separate conclusion summarizing essential findings and providing a clear overall research assessment.
Response: A separate section as “Conclusion” is added to the revised manuscript (Page 11).
Round 2
Reviewer 1 Report
Comments and Suggestions for Authors
The authors answered all questions.
Author Response
Thank you for your comments.
Reviewer 3 Report
Comments and Suggestions for Authors
Dear Authors
I am grateful for your correction in response to my earlier comments. Your efforts to resolve comments certainly improved the overall quality of the manuscript. However, it is essential to emphasize that there are still areas where further improvements could be helpful. I thank you for your tremendous commitment to improving the work,
Best Regards
Introduction:
Clear and concise: The introduction can benefit from being brief and focused. Some sentences are long and have multiple ideas, which can be difficult for readers to follow. Consider breaking down complex ideas into smaller, more digestible sentences.
Expanding on Points: Although the introduction discusses the definition and importance of empathy, compassion, and compassion fatigue, it would benefit from expanding on these points. Provide more context and detail to help readers better understand concepts.
Connection to research: Link the introduction to the research topic and the systematic review. Explain how empathy, compassion, and compassion fatigue relate to the review’s focus.
Avoid complacency: Some words seem to echo similar thoughts, such as the description of empathy and compassion. Take proper notes to eliminate redundancies and keep readers interested.
Provide a clear objective: Provide a clear purpose or objectives for the systematic review at the end of the introduction.
Methods:
If the review protocol has been approved, it should be made readily available on platforms such as the Open Science Framework (OSF). This transparency can enhance the credibility and replicability of the systematic review.
Results:
The reasons for excluding the 570 studies should be included in the PRISMA (Preferred Reporting Items for Systematic Reviews and Meta-Analyses) documentation.
Discussion:
Although the discussion provides a good analysis of the policy analysis and provides valuable insights, there are still areas for improvement:
Clarity and structure: Collaboration can benefit from a clear structure and structure. Breaking down articles into specific sections or topics makes it easier to follow the critique.
Provide more specific examples: The discussion can be enriched by including specific examples from reviewed studies. For example, highlighting studies in a systematic review and making specific references through their approach or findings adds depth to the critique.
Recommendations for future research: Although the discussion addresses the need for well-designed RCTs, it could go further by providing specific recommendations for future research. What should these RCTs focus on? Are there specific interventions or populations that need more attention?
Discussion of potential bias: While the discussion acknowledges that there is a high risk of bias in the included studies, it can explain the biases that were identified and their potential impact on the results and the details of the. This will provide a detailed assessment of study quality.
Additional instrument descriptions: When considering assessment instruments such as the ProQOL scale or salivary amylase, briefly describe what these instruments measure and why they are appropriate.
Discussion of ethical considerations: Because interventions may have had ethical implications for healthcare professionals, it would be useful to discuss ethical considerations associated with interventions in which they have been studied and a brief description of their systematic treatment
Author Response
We thank reviewer for their comments. It has definitely helped to make manuscript better.
Introduction:
- Comment: Clear and concise: The introduction can benefit from being brief and focused. Some sentences are long and have multiple ideas, which can be difficult for readers to follow. Consider breaking down complex ideas into smaller, more digestible sentences.
Response: We have edited few long sentences as advised (lines 47-50 & lines 62-67).
- Comment: Expanding on Points: Although the introduction discusses the definition and importance of empathy, compassion, and compassion fatigue, it would benefit from expanding on these points. Provide more context and detail to help readers better understand concepts.
Response: Introduction has text (line 29-32) to cover the definition and importance of empathy, compassion, and compassion fatigue as suggested...
“Driven by empathy, compassion is the deep feeling that arises after getting distressed by another person's suffering, accompanied by a strong desire to help and improve his/her wellbeing . Compassion leads to acts of kindness, taking empathy a step ahead. “
Please note that the other reviewers has commented that the ‘Introduction’ was too long, not focussed” and needed editing. Hence, we reduced the word count of this section from 580, to 480 words
- Comment: Connection to research: Link the introduction to the research topic and the systematic review. Explain how empathy, compassion, and compassion fatigue relate to the review’s focus.
Response: The definitions and the link between empathy, compassion and compassion fatigue has already been provided lines 29-32. Expanding on these issues any further will lead to redundancy.
- Comment: Avoid complacency: Some words seem to echo similar thoughts, such as the description of empathy and compassion. Take proper notes to eliminate redundancies and keep readers interested.
Response: We agree that empathy and compassion echo similar thoughts and many investigators use the terms interchangeably. However, academically, there is a difference between these entities. Please see below the lines reproduced from Dowling et al 2018. (Dowling T. Compassion does not fatigue! Can Vet J. 2018 Jul;59(7):749-750. PMID: 30026620; PMCID: PMC6005077. https://www.ncbi.nlm.nih.gov/pmc/articles/PMC6005077/ )
“Empathy is a mental construct that allows us to resonate with others’ positive and negative feelings. We can feel happy at the joy of others and we can feel distress when we observe someone in physical or mental pain. While sharing positive emotions with others is certainly pleasant, the sharing of negative emotions can be difficult.”
“In contrast to empathy, compassion is characterized by feelings of warmth, concern, and care for the other, as well as a strong motivation to improve the other’s wellbeing. Compassion goes beyond feeling with the other to feeling for the other. Unlike empathy, compassion increases activity in the areas of the brain involved in dopaminergic reward and oxytocin-related affiliative processes, and enhances positive emotions in response to adverse situations (8).”
Empathy and Compassion: What’s the Difference?
https://hbr.org/2021/12/connect-with-empathy-but-lead-with-compassion
We have carefully checked the text to avoid complacency as suggested and we believe no changes are required.
- Comment: Provide a clear objective: Provide a clear purpose or objectives for the systematic review at the end of the introduction.
Response: We have clarified this as suggested line 62-67.
“Given the significance of underlining issues, we aimed to review the current evidence on the effect of various strategies for reducing compassion fatigue in healthcare providers. Our specific objective was to conduct a comprehensive and robust systematic review fo-cused exclusively on randomised controlled trials (RCTs) of interventions for compassion fatigue in healthcare providers. Our findings will help in guiding research and clinical practice in the field”
Methods:
- Comment: If the review protocol has been approved, it should be made readily available on platforms such as the Open Science Framework (OSF). This transparency can enhance the credibility and replicability of the systematic review.
Response: The protocol was approved and registered by the Deakin University Supervisor as part 1 of the submission (Introduction, Aim, and Methods i.e., the protocol for the systematic review) towards the ‘Leadership Research project”. Part 2 was the entire manuscript ~ 7500 words including Part 1 and plus ‘Results, Discussion, and Conclusion with “Directions for further research”). Furthermore, this work was done during critical illness in the family of the first author, and there was a deadline to meet otherwise the entire semester would have been missed. Hope this clarifies the situation.
Retrospective registration will be inappropriate and unnecessarily caste a doubt on the integrity of the review.
Results:
- Comment: The reasons for excluding the 570 studies should be included in the PRISMA (Preferred Reporting Items for Systematic Reviews and Meta-Analyses) documentation.
Response: These 570 studies were excluded after reading the title and abstracts as they were not relevant to our systematic review as mentioned in PRISMA flow chart. This is the conventional approach of explaining the exclusion of studies.
Discussion:
- Comment: Although the discussion provides a good analysis of the policy analysis and provides valuable insights, there are still areas for improvement:
Response: We have carefully read the ‘Discussion’ and rechecked the comments by other two reviewers. We sincerely believe that except for few minor changes there is nothing that can be added to increase the value of this section of this very focused systematic review (as appreciated by other two reviewers).
- Comment: Clarity and structure: Collaboration can benefit from a clear structure and structure. Breaking down articles into specific sections or topics makes it easier to follow the critique.
Response: We appreciate the suggestion. However, we are unable to improvise the clarity and structure any further given the conventional format of reporting systematic reviews and approach to scientific manuscripts (Summarise overall key findings in a descriptive manner “without repeating the results”, what they mean, how they compare with previous studies, strength and limitations, directions for further research and data-based conclusion). Other two reviewers have not commented on this issue.
(10) Comment: Provide more specific examples: The discussion can be enriched by including specific examples from reviewed studies. For example, highlighting studies in a systematic review and making specific references through their approach or findings adds depth to the critique.
Response: Thanks for the suggestion. Other reviewers have not raised this issue. Considering the text under improvised ‘Discussion’ and the details provided on the characteristics of included studies (Table), risk of bias assessment (Figure), and added lines on “approach to exploration of heterogeneity” (as suggested by other reviewer) we believe that this issue has been already addressed adequately.
(11) Comment: Recommendations for future research: Although the discussion addresses the need for well-designed RCTs, it could go further by providing specific recommendations for future research. What should these RCTs focus on? Are there specific interventions or populations that need more attention?
Response: Given the significant heterogeneity in almost all aspects of the included RCTs, it is difficult to provide specific recommendations. We did provide options for the design of future trials, and the need to consider influence of confounders including setting, and culture in designing future studies. Please see the added text that addresses the comment (line 294-297).
“The first responders and those in intensive care set-up could be the priority population for future trials given they are at high risk of compassion fatigue. Equally important will be the considerations such as the settings (high vs low-income counties), language, and culture. Considering the ethical implications for any intervention to be assessed is an important issue in the design of future RCTs.”
- Comment: Discussion of potential bias: While the discussion acknowledges that there is a high risk of bias in the included studies, it can explain the biases that were identified and their potential impact on the results and the details of the. This will provide a detailed assessment of study quality.
Response: Thanks for the suggestion. Other reviewers have not raised this issue. Please see our response to comment 9 above. Considering the improvised ‘Discussion’, the results of the risk of bias assessment (figure) and added lines on “approach to exploration of heterogeneity” (as suggested by other reviewer) we believe this issue has been addressed adequately.
- Comment: Additional instrument descriptions: When considering assessment instruments such as the ProQOL scale or salivary amylase, briefly describe what these instruments measure and why they are appropriate.
Response: We have already provided details on ProQOL as a well-known tool to measure professional quality of life.
We have added text about salivary amylase to address this issue lines 294-297.
“Salivary amylase is a marker of stress-induced activation of the autonomic nervous system. Its secretion is controlled by sympathetic activation through β-adrenergic receptors. It has been as a marker to measure effects of stress reduction interventions.”
- Comment: Discussion of ethical considerations: Because interventions may have had ethical implications for healthcare professionals, it would be useful to discuss ethical considerations associated with interventions in which they have been studied and a brief description of their systematic treatment
Response: We have added a line to cover this point in the last line of the discussion line 300-301. A comprehensive discussion on this issue is beyond the scope of this focused systematic review.